# Shotgun Metagenomics-Guided Prediction Reveals the Metal Tolerance and Antibiotic Resistance of Microbes in Poly-Extreme Environments in the Danakil Depression, Afar Region

**DOI:** 10.3390/antibiotics12121697

**Published:** 2023-12-04

**Authors:** Ermias Sissay Balcha, Felipe Gómez, Mesfin Tafesse Gemeda, Fanuel Belayneh Bekele, Sewunet Abera, Barbara Cavalazzi, Adugna Abdi Woldesemayat

**Affiliations:** 1School of Medical Laboratory Science, College of Medicine and Health Sciences, Hawassa University, Hawassa P.O. Box 1560, Ethiopia; ermias.sisay@aastu.edu.et; 2Department of Biotechnology, College of Biological and Chemical Engineering, Addis Ababa Science and Technology University, Addis Ababa P.O. Box 16417, Ethiopia; mesfin.tafesse@aastu.edu.et; 3Centro de Astrobiología (INTA-CSIC) Crtera, Ajalvir km 4 Torrejón de Ardoz, P.O. Box 28850 Madrid, Spain; gomezgf@cab.inta-csic.es; 4School of Public Health, College of Medicine and Health Sciences, Hawassa University, Hawassa P.O. Box 1560, Ethiopia; fanuelbelayneh@hu.edu.et; 5Department of Microbial Ecology, Netherlands Institute of Ecology (NIOO-KNAW), P.O. Box 50, 6700 AB Wageningen, The Netherlands; s.aberadinke@nioo.knaw.nl; 6Institute of Biology, Leiden University, P.O. Box 9500, 2300 RA Leiden, The Netherlands; 7Ethiopian Institute of Agricultural Research (EIAR), Addis Ababa P.O. Box 2003, Ethiopia; 8Dipartimento di Scienze Biologiche, Geologiche e Ambientali, Università di Bologna, 40100 Bologna, Italy; barbara.cavalazzi@unibo.it; 9Department of Geology, University of Johannesburg, Johannesburg P.O. Box 524, South Africa

**Keywords:** antibiotic resistance genes, metal resistance genes, shotgun metagenomics, heavy metals

## Abstract

The occurrence and spread of antibiotic resistance genes (ARGs) in environmental microorganisms, particularly in poly-extremophilic bacteria, remain underexplored and have received limited attention. This study aims to investigate the prevalence of ARGs and metal resistance genes (MRGs) in shotgun metagenome sequences obtained from water and salt crust samples collected from Lake Afdera and the Assale salt plain in the Danakil Depression, northern Ethiopia. Potential ARGs were characterized by the comprehensive antibiotic research database (CARD), while MRGs were identified by using BacMetScan V.1.0. A total of 81 ARGs and 39 MRGs were identified at the sampling sites. We found a copA resistance gene for copper and the β-lactam encoding resistance genes were the most abundant the MRG and ARG in the study area. The abundance of MRGs is positively correlated with mercury (Hg) concentration, highlighting the importance of Hg in the selection of MRGs. Significant correlations also exist between heavy metals, Zn and Cd, and ARGs, which suggests that MRGs and ARGs can be co-selected in the environment contaminated by heavy metals. A network analysis revealed that MRGs formed a complex network with ARGs, primarily associated with β-lactams, aminoglycosides, and tetracyclines. This suggests potential co-selection mechanisms, posing concerns for both public health and ecological balance.

## 1. Introduction

Antibiotics represent a distinct class of therapeutic agents in the fight against infectious diseases [1]. They are produced by microorganisms and kill pathogenic microbes by targeting specific microbial components [1]. However, the emergence of antibiotic resistance poses a serious threat to the effectiveness of antibiotics, leading to a worldwide public health crisis that cannot be understated [2,3,4]. According to the World Health Organization’s (WHO’s) 2014 report, the world is at the brink of a post-antibiotic era, characterized by inoperable infections, an increased mortality rate, and escalating healthcare costs [5]. In 2019, an estimated 4.95 million deaths were attributed to antibiotic resistance, with 1.27 million cases directly caused by infections resulting from this phenomenon [4]. If effective preventive measures are not implemented immediately, the worldwide death toll from antibiotic resistance is projected to potentially surge to 10 million annually by 2050 [6]. Human activities, such as the improper use of antibiotics, inappropriate disposal of unused medicines, as well as intensive agricultural practices, have exacerbated this natural phenomenon. The increasing concern over the rise of antibiotic-resistant bacteria in clinical settings has presently prompted efforts to identify the environmental reservoirs of antibiotic resistance genes (ARGs), as these reservoirs serve as crucial intermediaries and important routes for the spread of antibiotic resistance [7,8,9]. Lakes, rivers, hot springs, and coastline areas, including soils, can act both as natural reservoirs of antibiotic resistance and pathways for the dissemination of clinically relevant ARGs [10]. 

While the research has primarily focused on antibiotic resistance in thermophilic and mesophilic bacteria, both pathogenic and non-pathogenic extremophilic bacteria have received less attention, despite their extensive use in biotechnological and industrial sectors. For instance, moderate thermophiles, which cause diseases, such as meningitis, endocarditis, and septicemia, have been found to be resistant to various antibiotics, including erythromycin, tetracycline, sulfamethoxazole, tobramycin, and netilmicin [11]. Similarly, isolates of *Arthrobacter* sp. and *Hafnia* sp. from hot springs have been found resistant to antibiotics [12]. However, the role of poly-extremophilic environments, which accommodates multiple extremities of temperature, salinity, and heavy metal concentrations, in the distribution of ARGs and their antibiotic resistance profiles remains underexplored. 

In addition to the biological factors, the widespread distribution of heavy metals in the environment is another consequential aspect of industrialization and urbanization that has had harmful impacts on human health and ecological risks [13]. Growing evidence suggests that metal contamination in natural environments significantly contributes to the persistence and proliferation of antibiotic resistance [14,15,16,17,18]. The historical exposure of bacteria to metals has made them a major source of environmental contamination. It is believed that co-selection is one of the primary mechanisms through which heavy metals contribute to the increased ARG levels in bacterial-rich environments [19,20,21,22,23]. For instance, soil bacterial resistomes have been found to expand under the selective pressure of Cu exposure [24]. Similarly, Hu and co-authors [25] reported an increase in the frequencies and abundances of ARGs in agricultural soils with elevated nickel levels. Recent studies using metagenomics and metagenome-assembled genomes have further demonstrated the extensive co-occurrence of ARGs and MRGs in bacterial hosts found within activated sludge and urban rivers [26,27]. This co-occurrence might suggest a significant environmental risk in areas with high levels of heavy metals. The close linkages between these resistance genes suggest the potential for co-transferability and co-expression mechanisms. This study was conducted in the Danakil Depression, specifically Lake Afdera and the Assale salt plain, where salt mining is a routine activity. The unregulated extractions of salt in these areas can lead to human contamination, as salt miners often lack the proper sanitation facilities. Moreover, Lake Afdera is utilized for both bathing and laundry, enhancing the potential for contamination. Consequently, it is imperative to ensure the health-related safety of these ecologically important reservoirs from further anthropogenic interventions. This is crucial not only because they can potentially serve as reservoirs for the spread of ARGs, but also due to their general exposure to elevated levels of heavy metals. The primary goal of this study is to determine the abundance of ARGs and MRGs in Lake Afdera and the Assale salt plain in the Danakil Depression. The study also aims to explore the potential correlations between environmentally toxic heavy metals and their associated ARGs. For this purpose, shotgun metagenome sequences are mined against databases, such as the comprehensive antibiotic resistance database (CARD) and BacMetScan V.1.0., available from the BacMet AntiBacterial Biocide and MRG databases to identify potential ARGs and MRGs, respectively.

## 2. Results

### 2.1. Physicochemical Analysis

Table 1 presents the physico-chemical measurements for water and salt crust samples collected from Lake Afdera (Figure 1) and the Assale salt plain (Figure 1). The Assale salt plain is extremely hypersaline (360 g/L) and more acidic (4.61) and warmer (39 °C) than Lake Afdera. Though both Lake Afdera (−111 m) and the Assale salt plain (−119 m) are notable for their low elevation points, the Assale salt plain is among low-land areas in Africa. The elemental composition exhibited a distinct variation among the two sample sites (Table 1). 

The concentrations of all measured metals were the highest in the Assale salt plain, except for Pb^2+^, which was slightly higher in Lake Afdera (Table 1). Concentrations of Mg^2+^ and Ca^2+^ ions in Afdera were almost half the amount of those found in the Assale salt plain. The heavy metal contents of Cu, Zn, and Fe in the Assale salt plain were by far higher than the measurements recorded in Lake Afdera. The measurements of other physicochemical parameters are listed in Table 1.

### 2.2. Metagenomic Data Analysis

Overall, the Illumina (Hong Kong, China) sequencing, using NovaSeqPE150 (Novogene, Hong Kong, China) generated an average of 6621 Mbp reads per sample with the GC contents determined to be 62% for the Assale salt plain and 56% for Lake Afdera. After quality trimming, the average cleaned datum was 6609.08 Mbp. From the assembled metagenomic reads, there were 47,042 and 93,220 scaffolds for the Assale salt plain and Lake Afdera, respectively. The scaffolds with average lengths of 569.5 bp were selected for the gene functional annotation. Table 2 presents the overview reads generated through the sequencing process. The data were assessed for the distribution of the base call quality, with % bases >Q20 found to be 97.72% and 96.02% for the Assale salt plain and Lake Afdera, respectively. 

The estimates of alpha diversity indices unveiled significant differences between Lake Afdera and the Assale salt plain. Both the Shannon and Simpson indices for bacterial communities demonstrated higher values for Afdera, and a lower value was obtained for the Assale salt plain (Table 3). These results suggest that Lake Afdera exhibits a higher level of diversity and evenness than the Assale salt plain. Nonetheless, the general diversity values for both Lake Afdera and the Assale salt plain were relatively low. 

### 2.3. Bacterial Community Profile

According to the phylum-wise distribution of the metagenomics data, Pseudomonadota was found to be the most dominant phylum in Lake Afdera, 63.61%, followed by Actinomycetota (13.95%), Bacilliota (7.82%), and Bacteroidota (2.61%) (Figure 2). Pseudomonadota, likewise, dominated in the Assale salt plain, accounting for 86.95% of all the organisms, with Bacilliota (3.49%), Cyanobacteria (1.98%), and Actinomycetota (0.012%) trailing behind (Figure 3). 

Distribution at the genus level displayed a distinct variation. Acinetobacter (18.57%) was found to be the most prevalent genus in Lake Afdera, followed by Pseudomonas (11.76%), Microbacterium (7.61%), Bacillus (4.69%), Methylobacterium (4.11%), and Rheinheimera (2.46%). However, in the Assale salt plain, Pseudomonas dominated significantly with 78.47%, trailed by Bacillus (3.33%), Aeromonas (0.241%), Rhodopseudomonas (0.11%), Rheinheimera (0.10%), and Methylobacterium (0.011%) (Figure 3).

### 2.4. Metagenomic Studies of ARGs

Through CARD, a total of 81 resistance genes associated with 16 antibiotics were identified in the metagenome of Afdera. Among these, those related to beta lactam (26), multidrug (21), aminoglycoside (9), and tetracycline (9) were present in higher abundances compared to other resistance genes (Figure 4, Appendix A). Some of the detected resistance genes included the fluoroquinolone class (*emrB*), β-lactamase (*ACC-1*, *OXA-58*, *OXA-363*, *OXA-212*, *OXA-50*, *NDM-17*, *OXA-134*, *ACT-29*, *LRA-19*, others), multidrug resistance (*abeM*; *abeS*; *mgrA*; *adeJ*; *MexC*; *adeL*; *adeH*; others), sulfonamide (*sul2*, *sul1*), tetracycline (*tet39*; *tetX*, *tetK*, others), and others (Appendix A). Only a few ARGs were identified in the Assale salt plain: β-lactamase class (*LRA-13*), diaminopyrimidine (dfrA10), and multidrug resistance efflux pump (adeF) (Appendix A). The findings indicate the presence of several ARGs in Lake Afdera compared to the Assale salt plain. 

For the Clusters of Orthologous Genes (COG) classification, various resistance genes associated with each antibiotic class are also described in Figure 4. The macrolide antibiotic class displayed a shared identity with COG predicted genes, like *mphD*, *macA*, and *srmB*; aminoglycoside aligned with COG predicted genes, such as *aadA24*, *APH6-Id*, *AAC6-IIc*, and others; tetracycline was associated with COG predicted genes *tet32*, *tetK*, *tetH*, *tetO*, and others; fluoroquinolone showed a similarity to COG predicted gene *emrB*; Beta Lactam was associated with COG predicted genes, such as *ACC-1*; *OXA-58*; *OXA-363*; *OXA-212*; and others; and streptogramin was associated with the predicted gene *vatD* (Appendix A).

The macrolide gene displayed the highest degree of identity (100%) with the Methylobacterium *mphD* gene (genes for macrolide phosphotransferase), whereas the fluoroquinolone gene shared the closest identity (99.43%) with the Enterobacteriaceae *emrB* gene. The gene for cephalosporin demonstrated the closest identity (98.02%) to Aeromonas (genes for *OXA* beta-lactamase, *OXA-212*). The Aminoglycoside gene exhibited the closest identity (82.8%) to *Acinetobacter baumanni* (genes for aminoglycoside nucleotide-transferase *ANT3-IIc*), while the tetracycline resistance gene (Tetracycline efflux gene *otrB*) revealed a 100% identity match to the *Acinetobacter lwoffii* (*otrB*) gene. The sulfonamide gene showed the closest identity match (100%) to the genes for sulfonamide resistance (*sul1*) from *Acinetobacter baumanni* and sulfonamide resistance (*sul2*) from Pseudomonodota phylum. In the same analogy, the ARGs identified from the Assale salt plain were fluoroquinolone and tetracycline resistance genes. The fluoroquinolone and tetracycline genes showed the highest degree of identity (98.97%) with the *Acinetobacter adef* gene.

### 2.5. Metagenomic Studies of MRGs

The BacMet2 database annotation identified a total of 39 resistance genes associated with eight heavy metals. These resistance genes comprised Cu, Zn, Pb, Cr, As, Hg, Cd and multi-metal resistance (involving Cu, Zn, Pb, Co, Mg, Hg, Cd, and Ni). Among the MRGs, Cu, Hg, and Cd were the most abundant in the overall dataset (Figure 5, Appendix A). Interestingly, the numbers of MRGs were noticeably higher in Lake Afdera than in the Assale salt plain (Figure 5, Appendix A). Copper resistance genes or resistance proteins, such as *CopC* and *CopD*, displayed a perfect match (100%) with the gene cluster from *Frankia* sp. *CcI3*. The cadmium tolerance gene, *CadD*, showed a 99.51% closest identity to *Staphylococcus aureus* A8819. Similarly, the Mercury resistance gene coding for Mercuric reductase (EC 1.16.1.1) was a complete match (100%) to the *Enterobacter cloacae* subsp. *cloacae ATCC 13047*. The chromate resistance protein (*ChrB*) and chromate transport protein (*ChrA*) showed 100% and 99.76% similarities to genes found in *Brucella tritici* and *Pseudomonas aeruginosa*, respectively. In addition, microbial communities in Lake Afdera possessed genes responsible for combined heavy metal resistance, such as nickel–cobalt–cadmium-tolerant genes (nickel–cobalt–cadmium resistance proteins *NccA* and *NccB*), exhibited the greatest identity match (99.25–99.91%) to *Cupriavidus pauculus*. Cobalt–zinc–cadmium resistance protein (*CzcD*) was a perfect match (100%) to *Pseudomonas aeruginosa*. The Cd(II)/Pb(II)-responsive transcriptional regulator had the closest identity to *Bacillus cereus* R309803. Magnesium and cobalt efflux proteins *CorC* had a 99.66% identity match with *Salmonella enterica* subsp. *enterica serovar Saintpaul* str. SARA29 (Appendix A). In the case of the Assale salt plain, the copper-translocating P-type ATPase (EC 3.6.3.4) gene showed a complete (100%) identity match with *Pyrococcus yayanosii CH1.* Additionally, copper resistance genes (*CopB* and *CopG* proteins) shared the closest identity (100%) with *Archaeoglobus fulgidus* DSM 4304 and *Escherichia coli* (strain K12), respectively. The arsenic tolerance gene, arsenite oxidase (*arsO*), had a 100% match with *Roseomonas cervicalis ATCC* 49957. P-type ATPase involved in Pb (II) resistance (PbrA) showed a 68% identity match with *Ralstonia metallidurans* bacterial species. Moreover, the mercuric resistance operon regulatory protein showed a perfect match (100%) to *Cupriavidus metallidurans* CH34. 

According to the RAST and COG identified genes, the metal tolerance gene classes for cadmium (Cd) were *czcB*, *czcc*, *czcd*, and *others*; mercury (Hg) tolerance was linked to genes classes of *MerC*, *MerE*, *MerR*, and *others*; arsenic (As) tolerance was associated with the *ArsO* gene; the lead (Pb) resistance gene was linked to *PbrA*; and chromium (Cr) tolerance was associated with the genes *ChrB* and *ChrA*, all found in Lake Afdera. On the other hand, in the Assale salt plain, MRGs were also detected for Cu with COG predicted genes, such as *CopC*, *CopD*, *CopG*, and *others*; Hg with genes, like *MerB*, *MerR*, and *MIR*; Cd tolerance was associated with *cadC*; and no As tolerant genes were detected (Appendix A). The findings reveal a greater diversity of metal resistance genes in Lake Afdera compared to the Assale salt plain. 

### 2.6. Co-Occurrence of Bacterial Hosts and Resistance Genes

The scaffolds were classified into 88 taxa, representing four phyla. Pseudomonadota (82.6%) was the dominant phyla, followed by Firmicutes (7.6%), Actinobacteria (4.3%), and Bacteroidetes (1.1%), which collectively enriched the identified ARGs and MRGs (Appendix A). A network analysis was conducted to uncover the co-occurrence patterns of bacteria–MRGs–ARGs. The graph highlights the high densities of Pseudomonadota and related resistance genes compared with networks featuring Actinomycetota, Bacilliota, and Bacteroidota (Figure 6). This suggests that Pseudomonadata are the primary carriers of resistance genes affected by co-selection. 

### 2.7. Correlations between MRGs and ARGs

From the samples in Lake Afdera, a strong positive correlation exists between specific metals, like cadmium and the abundance of ARGs (*r =* 0.98, *p* = 0.001), and between mercury and the abundance of MRGs (*r* = 0.88, *p*= 0.02) (Table 4). 

In the Assale salt plain, significant strong correlations were observed between ARGs with heavy metal zinc (*r =* 0.93, *p* = 0.001) and between mercury and MRGs (*r* = 0.83, *p* ≤ 0.04) (Table 5).

Additionally, correlations between MRGs and their corresponding ARGs using the mean abundance levels of resistant genes were computed. Table 6 reveals the existence of significant positive correlations between ARGs and MRGs at both sample sites (*p <* 0.05). 

To further elucidate the correlations between MRGs and their corresponding ARGs, a network analysis was constructed. The graph reveals the co-occurrence patterns among the relative abundances of MRGs, ARGs, and heavy metals. The resultant network consisted of 23 nodes (2 MRGs, 14 ARGs, and seven heavy metals) (Figure 7).

## 3. Discussion

Over the course of the antibiotic era, human health has shown remarkable improvements, leading some scientists to claim the end of infectious diseases [28]. However, the dramatic rise in antibiotic resistance among microorganisms has emerged as a growing global concern for human health [5]. Although environmental microbes, which are not disease-causing and are closely linked with antibiotic synthesis, have been largely overlooked, they are presently recognized to play a vital role in the evolution of antibiotic resistance. It is believed that pathogens are naturally vulnerable to antibiotics and do not inherently possess ARGs, but acquire them from environmental microbes. At present, the mechanism through which ARGs evolve and spread in environmental microorganisms remains unclear. Notably, every habitat, even habitats isolated for centuries or millennia, contains bacteria resistant to antibiotics and genes linked to resistance [29,30,31].

The research on antibiotic resistance has primarily focused on bacteria from thermophilic, mesophilic, and psychrophilic environments [32]. However, no attention has been given to the unique poly-extreme environments of the Danakil Depression, in the northern Afar region of Ethiopia. To address this gap, this study was designed to investigate the distribution of ARGs and MRGs in the extreme settings of Dallol and Lake Afdera. Previous studies have shown that the Danakil Depression is heavily enriched with various heavy metals, contributing to the remarkable color variations observed in the region’s hot springs [33,34,35]. The combination of high metal concentrations and extreme physico-chemical parameters (e.g., temperature, pH, and salinity) has led to the creation of a unique multi-extreme environment in the Danakil Depression. A taxonomic analysis revealed the dominance of Pseudomonadota in both extreme environments, suggesting a resemblance in the bacterial community structures between the two locations. While the abundance percentage varied, the major phyla, Pseudomonadota, Actinomycetota, and Bacilliota, were consistent, except for Bacteroidota, which was found exclusively in Lake Afdera and not in the Assale salt plain. This miniscule community composition difference might stem from variations in the physicochemical factors and distant geographical locations [32]. The low taxonomic diversity in Lake Afdera and the Assale salt plain aligned with the previous findings in other extreme hot springs, such as Sikkim, a northeastern state in India [36]. Additionally, similarities in community structure can also be observed when compared to the Rehai geothermal field located in China [37]. 

Our metagenomic analysis detected several putative ARGs in Lake Afdera, while the Assale salt plain had less (Appendix A). Of the identified ARGs, half were considered housekeeping genes and similarity searches using CARD returned no matches. These genes included *ACC-1*, *OXA-50*, *vatD*, *MexC*, *APH3-Ia*, *adeH*, *OpmB*, *ACT-29*, *vanHB*, *LRA-19*, *AAC6-Ib-cr*, *SMB-1*, *LRA-13*, *TEM-71*, *chrB*, *AER-1*, *IND-5*, *tetX*, *tetO*, *rmtB*, *APH2-IIa*, *MexD*, *tet32*, *emrB*, *mexN*, *OpmH*, *lnuA*, *MuxB*, *mexM*, *srmB*, *Sed-1*, *tetH*, *dfrA10*, *Erm38*, *OpmH*, *arr-4*, *ErmR*, *EXO-1*, *mexM*, and *tetA46*. An annotation procedure via RAST revealed their multi-functionality properties. For instance, the *emrB* gene encodes both DNA gyrase subunit A (EC 5.99.1.3) and DNA gyrase subunit B (EC 5.99.1.3), which are engaged in cell division and DNA replication. Another example is the *ChrB* gene, which encodes a chromate transport protein. Interestingly, these genes, along with metal-dependent hydrolases of the beta-lactamase superfamily I proteins and multidrug resistance efflux pumps, were present in the RAST MAGs of Bacilliota (Appendix A). However, strains, like *Bacillus cereus* and *Lysinibacillus fusiformis,* despite carrying these genes, showed susceptibility to antibiotics (unpublished data). This suggests that these genes may serve alternative metabolic roles, making them potentially inactive or hypothetical in antibiotic resistance. Numerous studies on *Bacillus* species and their isolates have consistently produced findings that indicate their susceptibility to various tested antibiotics. For instance, a study conducted by Coonrod and co-authors demonstrated that 49 *Bacillus* isolates were susceptible when tested against six different classes of antibiotics [38]. Investigations in Jordanian hot springs and other extreme environments, such as remote cave microbiomes, Antarctic marine waters, and pristine mountain rivers, have also shown the susceptibility of *Bacillus* strains to antibiotics [39]. The presence of resistance genes for β-lactam antibiotics is not surprising, given the involvement of β-lactamases in various common bacterial functions, like cell wall biosynthesis, signaling molecules, the detoxification of metabolites, and other processes [40]. 

The limited abundance of a few phyla, Actinomycetota and Bacilliota, which are not the key hosts of ARGs, combined with the observed low diversity, suggests diminished competition among microbial communities for acquiring ARGs in the current ecosystem. Moreover, the metagenomic investigation of the Assale salt plain identified only a few ARGs, predominantly related to Gram-negative bacteria (Appendix A). This scarcity of ARGs might be intrinsic to the environment or result from contamination, potentially from salt soil microflora or human skin flora introduced during salt mining practices. Previous research by Miller et al. emphasized the link between human activities and antibiotic resistance in remote sampling stations, such as Palmer in the Antarctic [41]. Given the uncertainty surrounding this issue, we examined the abundance of heavy metals and MRGs, and also the correlations that possibly existed between ARGs and MRGs. Several heavy metals were detected, including Cu, Fe, Zn, Cd, Ni, and Pb. The samples from the Danakil Depression, especially those from Lake Afdera, which originated from a depth of 160 m, were influenced by the surrounding rocks, leading to the wide distribution of metallic pollutants. Notably, Cu and Zn were abundant in the study sites and were directly linked to the development of heavy metal tolerance in the environmental microflora since the dawn of time [42]. In line with this, metal tolerance was assessed and the potential co-occurrence of heavy metal resistance with antibiotic resistance was explored. While metals, like copper and zinc, are essential nutrients supporting various physiological and cellular functions in microorganisms, they become toxic at high concentrations [43]. This necessitates the development of resistance mechanisms by microorganisms in response to prolonged exposure [44]. Accordingly, MRGs associated with copper, cobalt zinc, cadmium mercury, nickel, lead, arsenic, and chromium were identified in this study. Corroborating these findings, MRGs were also found in a separate study conducted on a functional metagenomic analysis [32]. Previous research has documented a significant increase in metal tolerance, particularly for copper, among several bacterial isolates, including, but not limited to, *E. coli* [45], and *S. enterica* [46], *G. thermoleovorans*, and *G. thermantarcticus* [47]. 

In the environmental samples, heavy metals were observed alongside antibiotic-resistant microbial communities [48]. More than half of the total microbial resistance genes detected in our study were linked to commonly consumed antibiotics, such as betalactam (32.1%), aminoglycoside (11.1%), and tetracycline (11.1%), as well as heavy metals, like Cu, Hg, and Cd. Similar findings were reported in previous abundance-based surveys [49,50]. This pattern suggests a potential correlation between the abundance profiles of ARGs and MRGs across different microbial phyla. Pseudomonadota (82.6%) emerged as the dominant phylum carrying ARGs and MRGs, followed by Bacilliota (7.6%), Actinomycetota (4.3%), and Bacteroidota (1.1%), indicating a potential risk of resistance gene dissemination. This distribution suggests a correlation between microbial communities, particularly those associated with humans, and the abundance of ARGs and MRGs (Figure 6). The long evolutionary periods, influenced by significant anthropogenic activities, like unhygienic salt mining practices, might facilitate the dissemination of genes, providing microorganisms with enhanced protection against harsh environmental conditions [51,52]. 

The observed abundance profiles of ARGs and MRGs within various genome phyla, combined with the selection pressure from a metallic-rich environment, raise our interest. Additionally, the recently reported abundance correlation in clinically important genera (e.g., *Escherichia*, *Shigella*, and *Klebsiella*) [53] prompted us to investigate the possible co-occurrence between ARGs and MRGs. The Spearman’s correlation analysis (*r*^2^ ≥ 0.8, *p* < 0.05) revealed significant associations between specific heavy metals (Cd and Zn) and ARGs (Table 4 and Table 5). These findings are consistent with previous studies that also report similar correlations, particularly between Zn and ARGs originating from sulfonamides [53] and tetracycline [48], though the trends are diversified [54,55]. Additionally, significant correlations were observed between ARGs and MRGs (Table 6), as well as between specific ARGs, MRGs, and heavy metals from the network analysis (Figure 7). This supports the hypothesis of the co-occurrence of MRGs and ARGs in the resident microbes in the sampling sites, a phenomenon increasingly reported in the literature, particularly in intestinal microbes subjected to antibiotic and metal pressures [56,57,58]. Further, this co-occurrence was observed in experiments conducted by Pal and colleagues [50], suggesting a novel co-selection potential, and was also observed in full-scale biogas reactors [59]. Some other previous studies have also reported the selection of ARGs by various metals [48,60], including those found in animal manure [61] and copper tailing dam areas [55].

The co-selection of MRGs with ARGs is hypothesized to be primary pathway for the spread and persistence of antibiotic resistance in different environments [62,63]. This link can occur through shared functions, the co-regulation of gene expression, and physical co-localization of resistance genes (with MRGs and ARGs located in the same mobile element) [64]. Efflux pumps, like the multidrug efflux pump *mdrL* in *Listeria monocytogenes*, can lead to cross resistance by expelling both antibiotics and heavy metals [65]. In the cases of co-regulation, the gene expressions of both ARGs and MRGs were influenced by a common factor. For example, in *Pseudomonas aeruginosa,* the characterization of efflux pumps (*CzcCBA*), which are responsible for resistance to Zn and Cd, revealed that their expressions were regulated by two genes, *CzcS* and *CzcR*. These genes also controlled the expression of the *OprD* porin, which was responsible for resistance against carbapenems (a class of β-lactam antibiotics) [60]. Additionally, the co-localization of MRGs and ARGs on mobile genetic elements played significant role in co-selection, leading to the dissemination of ARGs as a consequence of heavy metal contamination. This suggests that ARGs can be maintained in the environment through the co-selection with MRGs in polluted areas, where the level of heavy metal contamination is remarkably higher than that of antibiotics pollution [66]. Overall, the results of the present study imply the heavy metal-induced selection of ARGs in multi-metal polluted extreme environments, raising concerns for human and animal health due to frequent exposure to the sampling area. Further studies are necessary to investigate the potential public health risks associated with these co-selection mechanisms and their role in promoting resistance in bacterial communities.

## 4. Materials and Methods

### 4.1. Study Area and Sampling Sites 

The study samples were collected from the hypersaline Lake of Afdera and Assale salt plain, both located in the geologically actives areas of the northern Afar Depression (Ethiopia) in the Great Rift Valley (Figure 1). The northern Afar Depression, which includes the Danakil Depression, is an incipient seafloor-spreading center located between the western scarp of the Ethiopian Plateau, the Danakil Alps to the east, and the Erta’Ale Range (NNW–SSE axial volcanic complex) to the south. The Danakil Depression is an arid, elongated (approximately NNW to SSE) lowland plain (~250 km long) mostly lying ~100 m below sea level (b.s.l.) in the northern part of Ethiopia. A significant portion of the Danakil Depression hosts vast evaporite deposits, covering an area of 4000 km^2^, known as the Assale salt plain. Both the Assale salt plain and Lake Afdera have been traditionally exploited for table salt extraction for human and cattle consumption for many years. The annual production ranges from 35,000 to 1.3 million tons [67]. 

The Assale salt plain near the Dallol geothermal area is characterized by a mix of extreme physico-chemical parameters, which include high temperatures, an acidic pH, hypersalinity, and the presence of high concentrations of heavy metals (e.g., iron: 35.6 g/L, copper: 93 mg/L, and zinc: 72 mg/L) [33,68]. In these extreme ecosystems, extremophilic organisms known as poly-extremophiles create selective pressure that favors their survival. These organisms have unique adaptive strategies, primarily involving heavy metal tolerance [33,34]. Lake Afdera is located in the Afdera Woreda in the southern part of Danakil Depression. The lake reaches a maximum recorded depth of approximately 80 m, with a total volume of around 2.4 km^3^ and reaching up to 112 m b.s.l. Its high salinity and distinct physicochemical parameters create an ideal environment for extremophiles. Lake Afdera is also enriched with various heavy metals, including lithium, which is of interest for industrial applications [69,70]. Sampling was performed in April 2021. We were only allowed to take samples from the water (2000 mL) of the lake, not the salt crust, by the local inhabitants. The community harbors reservations about sampling the salt crust due to the fear that such research can potentially cast salt extraction in a negative light, harming the reputation of the community’s primary means of livelihood. Physicochemical measurements (Table 1) were performed onsite using a portable refractometer (HI-9829-02 advanced portable Multi-parameter pH/ISE/EC/DO/Turbidity waterproof Meter, Eden Way, UK). GPS coordinates were recorded using GPSMAP64 (Garmin, Lenexa, Kansas, USA) as per the guidelines of the manufacturer (Table 1). The samples were collected randomly under aseptic conditions in sterilized thermal flasks (MegaSlim, Hawaii, USA) in triplicates considering minimal human and animal contact to reduce possible contamination. Prior to DNA extraction, the samples were transferred to the laboratory and kept at 4 °C and then processed immediately. 

### 4.2. Determining Metal Compositions 

The metal composition of the lake was determined by inductively coupled plasma optical emission spectrometer (ICP-OES), specifically the Agilent 5100 SVDV ICP-OES (Santa Clara, CA, USA), conforming to ES ISO 11885:2007 standards. The samples (50 mL) were subjected to digestion at 80 °C with 10 mL of nitric acid and then cooled, filtrated, and diluted to 100 mL with distilled water. The detection limit was 0.01 μg L^−1^. The standard working parameters were selected, and the procedures outlined by Van de Wiel were followed [71]. 

### 4.3. Metagenomic DNA Extractions and Shotgun Sequencing

DNA extractions were accomplished using modified 1% CTAB-SDS method adopted from Zhou and co-authors [72] at the molecular biology laboratory of Addis Ababa Science and Technology University. All the DNA extractions were performed in triplicates and replicates from each sample site were later pooled prior to metagenome sequencing. The quality and quantity of the extracted DNA was checked using Thermo Scientific NanoDrop 3300 Fluorospectrometer (Thermofisher Scientific, Wilmington, DE, USA). The DNA was randomly sheared into small fragments. These fragments were then end repaired and ligated into Illumina adapters. Sheared fragments underwent size-selection, PCR amplification and purification. Metagenome library was prepared per effective library concentration and the required data volume. The library was examined using ThermoFisher Qubit fluorometry, real time PCR quantitation and bioanalyzer to identify the size distribution. The library was barcoded, pooled and shotgun sequenced on one lane of a flow cell using a 150 bp paired-end run on a NovaSeq PE150 instrument (Illumina, Tsim Sha Tsui, Hong Kong). Sequencing reads were de-multiplexed using Cassava v.2.0 [73], FastQC v0.11.6 [74] was used to examine the quality of the paired-end raw reads, TRIMMOMATIC v0.36 [75] (Q-value ≤ 38; N > 10bp; reads overlap with adapter > 15bp) was used to remove any adapters contamination and low quality reads. 

### 4.4. Assembly, Binning, and Annotation of Reads

To generate the assembly of the reads, MEGAHIT [76] was used initially and then the assembly reads were mapped using Bowtie2 [77]. Assemblies were scaffolded to obtain the best contigs. Quast was used to compute the assembly statistics [78]. Using the technique outlined by [79], the assembled contigs were further binned using MetaBAT2 [80], CONCOCT [81], and MaxBin [82], and the retrieved metagenome assembled genomes (MAGs) were pooled further with DAS Tool (v1.1.1) [83]. CheckM was used to assess the quality of MAGs (≥80% completeness and ≤10% contamination) [84]. The metagenome constructed contigs were subjected to annotations using BLASTn [85] against the NCBI GenBank annotation pipeline and Rapid Annotation using Subsystem Technology (RAST) tools with the annotation scheme of Classic RAST [86]. The predicted genes were functionally classified based on the evolutionary genealogy of genes (egg) [87]; orthology was conducted using COG [88].

### 4.5. Taxonomic Assignment of Contigs 

The taxonomic profiling of the assembled contigs was conducted using the MEtaGenome ANalyzer 4 (MEGAN4) [89]. To taxonomically label each metagenomic homolog, MEGAN4 used sequence or phylogenetic similarities to the microNR database [90]. This was done by placing reads that were homologous to marker genes within a database of taxonomically informative gene families. 

### 4.6. Extracting Antibiotic and Metal Resistance Genes 

The CARD [91] was used to identify and characterize putative ARGs. All genes were subjected to a blastp (e-value ≤ 1 × 10^−5^) analysis against the CARD database [92,93]. Each individual gene was comprehensively annotated, including the resistance profiles and underlying resistance mechanisms. The relative abundances of the various resistance genes were also calculated [94]. In addition, the BacMetScan V.1.0 [94] script, which was accessible in BacMet AntiBacterial Biocide and MRG databases, was used to identify potential metal resistance genes. BacMet-Scan used a manually curated library of genes with resistance functions that were empirically confirmed [95]. Furthermore, a COG classification was also performed to predict resistance gene classifications using BLASTx against the COG database [88]. 

### 4.7. Statistical Analysis

The datasets were analyzed using packages in the R environment. Alpha diversity indices, such as Shannon’s and Sampson’s, were estimated by considering the abundance of each genus using an analysis of variance (ANOVA), and *p* < 0.05 values were considered significant. The ANOVA was performed by statistical PAST software version 4.14 [96]. Additionally, one-way ANOVA was used in SPSS version 26 to assess the differences of heavy metals, MRGs, and ARGs across the study sites. Spearman’s correlation analyses were also performed to identify significant correlations between heavy metal concentrations and resistant gene (MRGs and ARGs) abundances. Furthermore, a network analysis was employed to explore the co-occurrence patterns of MRGs, ARGs, and heavy metals. Statistically robust correlations having Spearman’s coefficients *>* 0.4 and *p*-values *<* 0.05 were taken as significant, and R software version 4.3.1 was utilized. A network analysis was conducted and visualized with i graph on the R package.

## 5. Conclusions

The persistence and proliferation of antibiotic resistance pose sever threats to humanity. Metal-polluted environments have been identified as natural reservoirs for clinically relevant ARGs. However, the origin and evolution of resistance genes remain poorly understood and require further extensive research. This study explored the effects of heavy metals on the proliferation of antimicrobial resistance in two extreme environments, Lake Afdera and the Assale salt plain, in the Danakil Depression in Ethiopia. A metagenomic approach was employed to study microbial diversity and understand the profiles of ARGs and MRGs in the study settings. The abundance of MRGs was positively correlated with Hg concentration, indicating that Hg played an essential role in the selection of MRGs. As for ARGs, their abundances were significantly correlated with the concentrations of Zn and Cd, suggesting that Zn and Cd may have induced antibiotic resistance in the sampling area. The network analysis revealed the co-occurrence patterns of heavy metals, ARGs, and MRGs among the resident bacteria communities. The co-selection of ARGs exhibited a preference for certain bacterial communities, particularly Pseudomonadota, potentially driving the proliferation of resistance genes. Among the various heavy metals studied, Cu and Hg triggered wider responses of resistance genes under a high selective pressure. The present research underscores the urgency of limiting heavy metal contamination in Lake Afdera and the Assale salt plain, as these regions are primary sites for the production of consumable commercial salts, and contamination can exacerbate antimicrobial resistance issues due to the co-selection effect on ARGs. Future studies should thoroughly examine bacterial resistance phenotypes to establish a comprehensive understanding of the genotype’s co-selection potential. Additionally, it is important to investigate the affiliated genes located near the co-existing ARGs and MRGs to understand other genetic variables that contribute to the co-selection of antibiotic and metal resistance. Therefore, it is necessary to identify all the potential co-selection agents and their involvement in the propagation of antibiotic resistance dissemination in environments associated with humans. This will improve the risk assessment of antibiotic resistance within the current clinical or environmental frameworks.

## Figures and Tables

**Figure 1 antibiotics-12-01697-f001:**
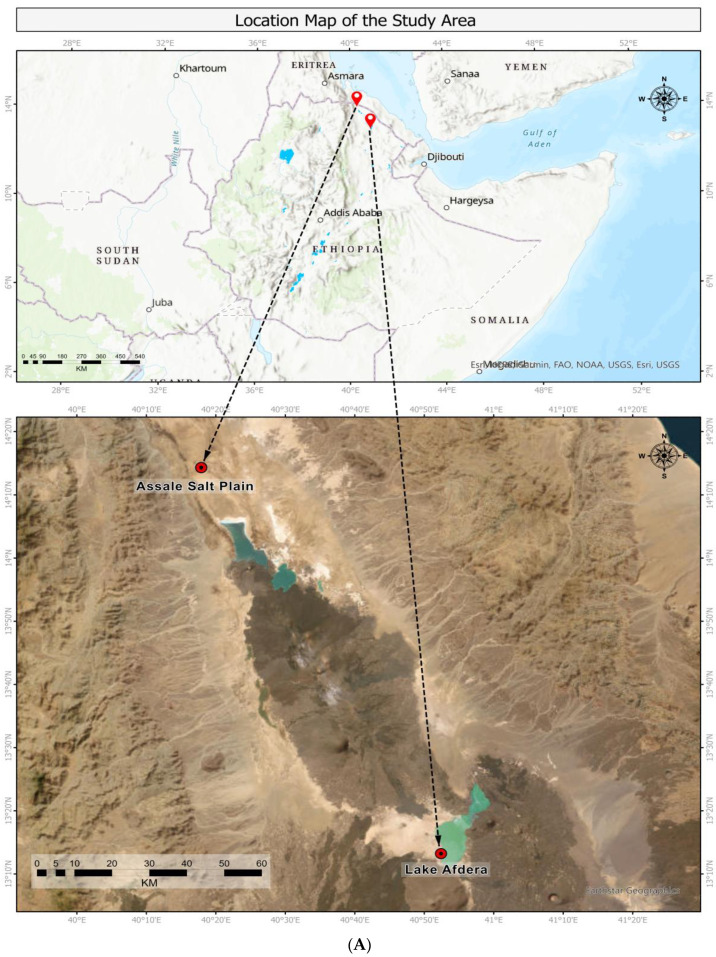
(**A**) Satellite image of the Lake Afdera and the Assale salt plain in northern Ethiopia. (**B**) and (**C**): panoramic views of the sampling sites of the Assale salt plain near the Dallol geothermal area and Lake Afdera, respectively.

**Figure 2 antibiotics-12-01697-f002:**
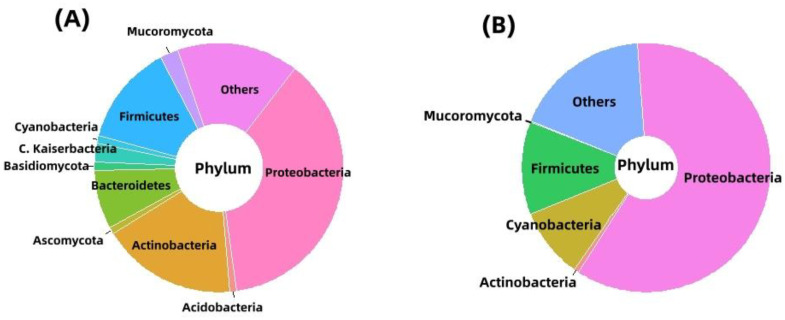
Bacterial community profile at the phylum-level diversity for (**A**) Lake Afdera and (**B**) Assale salt plain.

**Figure 3 antibiotics-12-01697-f003:**
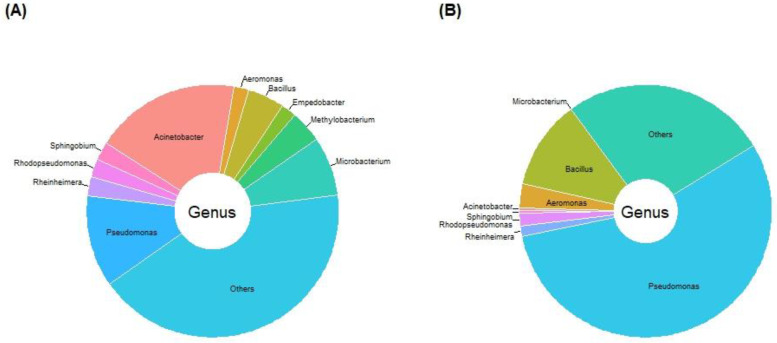
Bacterial community profile at the genus-level diversity for (**A**) Lake Afdera and (**B**) Assale salt plain.

**Figure 4 antibiotics-12-01697-f004:**
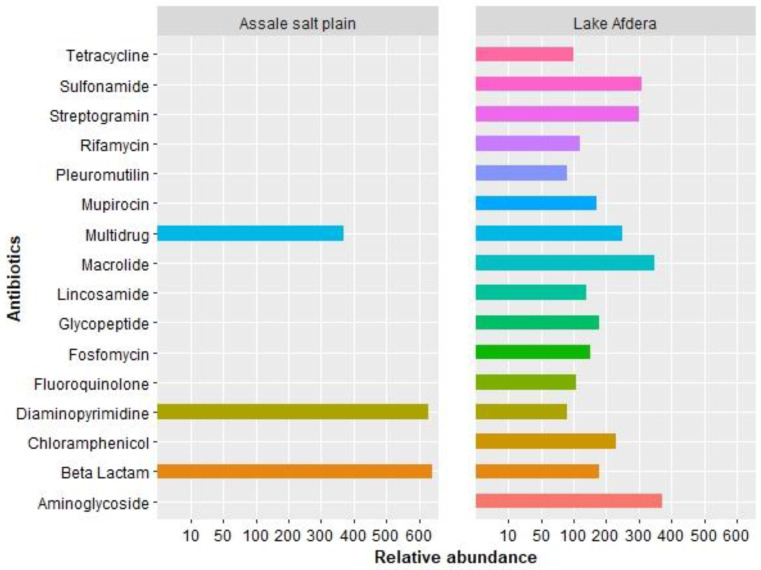
The abundance of antibiotic resistance genes (ARGs) in Lake Afdera and the Assale salt plain predicted using CARD, and a resistant gene classification was conducted based on the Clusters of Orthologous Genes (COG) classification.

**Figure 5 antibiotics-12-01697-f005:**
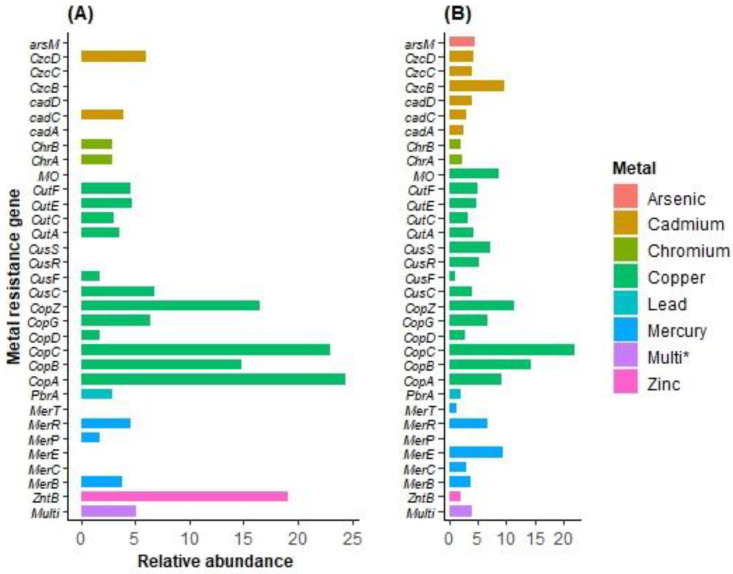
Abundance of metal resistance genes (MRGs) for the (**A**) Assale salt plain and (**B**) Lake Afdera identified from the BacMet AntiBacterial Biocide and MRGs database. Multi* refers to multi-metal resistance genes.

**Figure 6 antibiotics-12-01697-f006:**
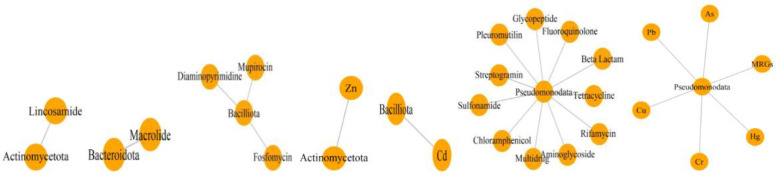
Co-occurrence patterns of potential bacterial hosts and their harbored resistance genes. The network analysis reveals profiles of bacterial hosts carrying both ARGs and MRGs in the four dominant phyla (Pseudomonadota, Actinomycetota, Bacilliota, and Bacteroidota).

**Figure 7 antibiotics-12-01697-f007:**
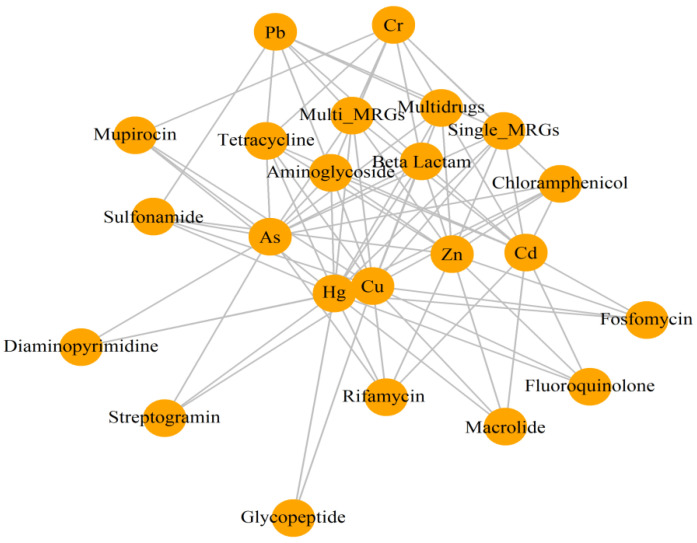
Co-occurrence networks of MRGs, ARGs, and heavy metals. The network was constructed using the igraph in the R interface, based on the observed co-occurrence patterns of MRGs, ARGs, and heavy metals in metagenomically sequenced bacterial genomes.

**Table 1 antibiotics-12-01697-t001:** Physicochemical data of the sampling sites (concentrations in g/L, T: °C, salinity: g/L, elevation: m).

Measured Parameters	Assale Salt Plain	Lake Afdera
Coordinates	0640063 E/1574494 N	0703016 E/1462263 N
pH	4.61	6.29
Temperature (°C)	39	36
Salinity (g/L)	360	137
Elevation (m)	−119	−111
Na (g/L)	49	38.5
Ca (g/L)	21.5	13.9
Mg (g/L)	1.98	1.1
Cl (g/L)	1.7	1.6
Mn (g/L)	0.12	0.618
Cu (g/L)	0.012	0.008
Pb (g/L)	0.0001	0.0002
Zn (g/L)	0.007	0.0001
Ni (g/L)	0.002	0.001
Fe (g/L)	0.049	0.002
Cd (g/L)	0.0001	0.00001

**Table 2 antibiotics-12-01697-t002:** Overview of metagenomics dataset for the Assale salt plain and Lake Afdera.

Parameters	Assale Salt Plain	Lake Afdera
Library insert size (bp)	350	350
Length of single read (bp)	150	150
Average raw reads (Mbp)	6479	6762
Total number of cleaned reads (Mbp)	6459	6759
GC content (%)	62	56
No. of contigs	47,042	93,220
Longest contig length (bp)	16,520	176,724
N50 (bp)	652	1212

**Table 3 antibiotics-12-01697-t003:** Alpha diversity index estimations.

Sampling Sites	Shannon	Simpson	*p*-Value
Lake Afdera	4.20	0.89	0.01
Assale salt plain	0.39	0.10	0.04

**Table 4 antibiotics-12-01697-t004:** Correlation between metal concentrations in the Lake Afdera sample and resistant gene abundance.

Metals	Metal Concentration	Correlation Coefficient			
		ARG Abundance	*p*-Value	MRG Abundance	*p*-Value
Cu	11.49	−0.08	0.87	0.54	0.26
Hg	0.17	−0.66	0.16	0.88	0.02
Cd	0.13	0.98	0.001	−0.12	0.82
Zn	7.16	−0.60	0.21	−0.03	0.96

**Table 5 antibiotics-12-01697-t005:** Correlation between metal concentrations in the Assale salt plain sample and resistant gene abundance.

Metals	Metal Concentration	Correlation Coefficient			
		ARG Abundance	*p*-Value	MRG Abundance	*p*-Value
Cu	0.88	−0.54	0.26	0.31	0.54
Hg	1.57	−0.31	0.54	0.83	0.04
Cd	0.02	0.08	0.87	−0.37	0.47
Zn	0.07	0.93	0.001	−0.32	0.53

**Table 6 antibiotics-12-01697-t006:** Correlation between abundance levels of ARGs and MRGs in Lake Afdera and the Assale salt plain samples collected in April 2021.

Samples	ARG Abundance Mean (±SD)	MRG Abundance Mean (±SD)	Correlation Coefficient	*p*-Value
Lake Afdera	25.02 (11.64)	4.91 (3.70)	0.44	0.033
Assale salt plain	2949.94 (1661.09)	8.76(7.42)	0.56	0.004

## Data Availability

The raw metagenomic reads were deposited in the Sequence Read Archive (SRA), NCBI, and Bio-Sample and SRA accession numbers were received. The accession number SAMN31412140 corresponds to the Assale salt plain with sample name “environmental metagenome”. Additionally, accession number PRJNA895852 represents Lake Afdera with the sample name “hypersaline lakes”.

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
