# Peer review of "Shotgun Metagenomics-Guided Prediction Reveals the Metal Tolerance and Antibiotic Resistance of Microbes in Poly-Extreme Environments in the Danakil Depression, Afar Region"

_antibiotics, 2023, doi:10.3390/antibiotics12121697_

Round 1

Reviewer 1 Report

Comments and Suggestions for Authors

The manuscript by Balcha et al reveals interesting findings on antibiotic resistance and metal resistance genes in extreme environments. The manuscript is well written. It is recommended that authors be more specific to the challenges on limited research from extremophilic environment.  

Some specific comments are as follows:

Abstract line 21-22: the explanation limited research is vague.

Line 47-48: Sentence is redundant antibiotic resistant/resistance is repeated. Please rephrase

Line 116-117: The values can be converted to g/L. Same applies to Table 1. 

Lines 120-125: Instead of rewriting the table values, the authors are advised to describe the difference as fold change.

Line 124: Elaborate the other parameters

Lines 147-154: Did the authors compare the alpha diversity indices by pairwise t-tests or ANOVA? The authors mention significant variations at what confidence interval? p values should be given.

Lines 166-172: Italicise all the genus names

Lines 182-192: Italicise all the gene names. Maintain this throughout the manuscript

Comments on the Quality of English Language

English is fine. Minor editing needed

Reviewer 2 Report

Comments and Suggestions for Authors

Overall the manuscript is good. Here are some suggestions.

1.       More explanation about Lake of Afdera and Assale salt plain on behalf of public health should increase the significance of the study site.

2.       The occurrence of ARG and MRG might be Co-resistance and Cross-resistance: Studies should discuss more on how exposure to heavy metals can select for bacteria For co-resistance mechanisms (where one gene provides resistance to both). And for cross-resistance (where resistance to one leads to resistance to the other) can also happen.

3.       According to the raw data, did the researchers find mobile elements contained both ARG and MRG.

4.       In line 300, fig 6 should be fig 7.

5.       Some improper word size (line 97-100)

Reviewer 3 Report

Comments and Suggestions for Authors

The authors report on a metagenomic analysis of bacterial samples from Lake Afdera and the Assale Salt Plain. They identify the predominant bacteria in each location,  and their associated antibiotic- (ARGs) and metal resistance genes (MRGs), as well as the metal concentrations in both environments. Correlations are established between various metals, and both kinds of resistance genes. This is an interesting paper that adds to the body of evidence indicating co-selection of MRGs with ARGs.

While the paper is well written, I strongly recommend that it be shortened, particularly in the discussion section. There is a lot of repetition here, from both the introduction and results sections, as well as some not-completely-relevant information. The conclusions also should be trimmed substantially.

Line 132: In the results, between sections 2.1 and 2.2, a description of what samples were collected, how and form where , is missing here. The relevant information is mostly somewhere in the materials and methods section, but some reference to what was collected and how, should be presented in the results.

Lines 197-205; is it really necessary, or even helpful, to list information already presented in tables and figures? Similarly, the lists in lines 240-250 are not really helpful.

Line 300: I think Figure 7 is what is meant here, not 6??
